# The Effect of WhatsApp Usage on Employee Innovative Performance at the Workplace: Perspective from the Stressor–Strain–Outcome Model

**DOI:** 10.3390/bs12110456

**Published:** 2022-11-16

**Authors:** Nur Muneerah Kasim, Muhammad Ashraf Fauzi, Muhammad Fakhrul Yusuf, Walton Wider

**Affiliations:** 1Faculty of Industrial Management, University Malaysia of Pahang (UMP), Gambang 26300, Pahang, Malaysia; 2Faculty of Business and Communications, INTI International University, Nilai 71800, Negeri Sembilan, Malaysia

**Keywords:** social media, information overload, communication overload, social overload, technostress, innovative job performance

## Abstract

Social media applications have increasingly become a valuable platform for personal communication and knowledge sharing in working life. Several researchers have considered the direct role of social media usage in influencing job performance. However, limited studies explore how social media use may impact employees’ job performance, especially in innovativeness. Moreover, inconsistencies in the findings exist in the literature regarding whether social media improves employees’ job performance or causes harm. By adapting the stressor–strain–outcome (SSO) model, the present study investigates how WhatsApp use at work can predict social media overloads that might induce technostress and, subsequently, affect employees’ innovative job performance. Thus, 206 Malaysian employees from the government and private sectors participated in this study and the data were analyzed using partial least squares structural equation modeling (PLS-SEM). The findings show that social media, predominantly WhatsApp, used at work has a mild but statistically significant influence on information overload, communication overload, and social overload. In addition, information overload and communication overload positively influence technostress, except for social overload. Subsequently, technostress does not have an impact on innovative job performance. This study provides theoretical and practical implications for extending the knowledge and mitigating plans and efforts to improve employees’ performance at work. Therefore, this study helps mitigate the dearth of research pertaining to the roles of social media use at work on employees’ innovative job performance.

## 1. Introduction

Social media has countless users worldwide and the number is constantly growing. The definition of social media varies [1]; according to [2], the definitions of social media presented in the literature and the commonalities among current social media services are: (1) social media services are currently Web 2.0 Internet-based applications; (2) user-generated content is the lifeblood of social media; (3) individuals and groups create user-specific profiles in an app designed and maintained by a social media service; and (4) social media services facilitate the development of social networks online by connecting a profile with those of other individuals or/and groups. Social media functionalities are not only traditionally designed for social networking purposes, but are also widely used for business and work purposes. Hence, many available social media platforms are widely used by organizations for official purposes, including Facebook, WeChat, DingTalk, WhatsApp, Twitter, blogs, YouTube, and photo-sharing sites [3,4].

Social media usage at work is regarded as a form of computer-mediated communication adopted by employees for work-related purposes [4,5], personal use [6,7] or both [8,9]. The rising trend of social media use at work has influenced employees to connect with social media, as it integrates with the routine activities of employees’ lives that directly affect their behavior [10]. Abundant research has discovered that social media use at work could enhance individual job performance, job satisfaction, job productivity, and work engagement as well as strengthen and maintain professional networks within or outside organizations [11,12,13,14].

Despite the potential benefits of using social media, scholars from several disciplines, including health, psychology, and computer–human behavior, have started to recognize the need to understand the potentially harmful and unintended consequences of social media usage in the workplace. As such, [15] mentioned that technology usage could have negative effects once it exceeds optimal-level usage. Individuals who continuously use social media tend to suffer from social media overload [16,17]. With the growing number of social media users and their activity levels, a large volume of information and communication can be generated that requires users to process it, which indirectly leads to the issues of overload on social media users [18]. This unpleasant condition is more likely regarded as a major techno-stressor that could negatively impact employees’ job performance [7,19]. Hence, there is a phenomenon of social media overload becoming more common, in parallel with social media growth.

Considering social media overload as the antecedent of the adverse outcomes of social media usage, the literature on social media overload has shown the indirect effect on technostress, social media fatigue, and social media exhaustion that subsequently lead to adverse outcomes, such as poor academic performance [20,21], discontinuous usage intention [4,22], and psychological issues [23,24,25,26]. Moreover, [27] found that employees who experienced social media overload suffered from social media exhaustion, leading to low job performance. Undoubtedly, social media overload can act as a specific stressor for technology use that induces strain, leading to adverse behavioral, psychological, and physiological outcomes.

Despite the existing body of knowledge, our understanding of social media overload is still constrained by some persisting gaps in the social media literature. The potential work-related consequences of social media stressors, especially in innovative job performance, remain understudied [19,22]. Most prior studies on social media overload concentrated more on general social media users [28,29,30] and students [31,32] rather than employees. Furthermore, studies on the different dimensions of overload remain scarce [30]. In the context of social media, the technostress associated with social media use has been studied primarily through the consequences of behavioral and psychological response [33,34,35,36] and little attention, to date, has been paid to the potential work-related outcomes, such as innovative job performance. Thus, this study provides a detailed investigation into how the association between social media use at work and social media overload can induce psychological strain that interferes with employees’ innovative job performance. The study focuses specifically on the use of WhatsApp in the Malaysian context.

The growing number of social media users, especially WhatsApp users, among employees has led to the phenomenon of social media overload. A large volume of information, communication, and social interaction may be generated from personal or work purposes, or both, which requires employees to process this information indirectly and excessively. Facing the same problems as other emerging technologies, social media use in the workplace has become contentious [9]. In addition, previous studies have shown inconsistent or mixed findings regarding whether social media use at work can increase or hinder employees’ job performance [1,3,27,37]. This study argues that this is a critical gap, because personal social media and smartphone use have significantly increased during working hours [38]. In consequence, employees cannot cope with this stressful situation, which may induce employee strain that turns into technostress. Hence, this study explores this undesirable situation by adapting the stressor–strain–outcome (SSO) model. Through the SSO model, the relationships between the different dimensions of social media overload are tested for their influence on technostress and, subsequently, the impact of social media overload on employees’ innovative job performance.

## 2. Literature Review

### 2.1. Underpinning Theory

The SSO model was initially developed by [39] to explain the stress process by determining how different stressors indirectly affect behavioral outcomes. The model has been applied to similar principles and concepts based on the transactional view of stress [30]. In addition, the SSO model has been extensively used in social media research to examine stress-related conditions, among other outcomes [16,20,30,40,41]. Basically, the model consists of three key aspects, namely (a) stressor, (b) strain, and (c) outcomes. The sequential process in the SSO model shows that the stressor will indirectly affect the strain and subsequently produce a specific outcome.

Within the SSO model, the stressor can be conceptualized as the environmental stimulus (emotional and behavioral), considered problematic [42], which appears irksome, annoying, or disruptive to social media users [16]. Meanwhile, the strain most often occurs due to the imbalance between the person’s situation and environmental demands [20]. The strain is defined as the psychological outcomes or the adverse emotion that negatively reacts to the environmental stimulus and mediates the effects of stressors on the specific outcomes [21]. Further, the outcome refers to a decrement in physical, behavioral, and psychological functioning, productivity as well as performance of an individual due to strain [42,43]. The SSO model is employed by considering social media overloads as stressors that induce employees’ strain, which refers to technostress, subsequently affecting their outcomes (innovative job performance).

Even though other theoretical models, such as the stimulus–organism–response or the transaction theory of stress, have been adopted in social media research, this study chooses the SSO model as a theoretical framework due to the underlying principles of the SSO model that are in line with the majority of technostress research [30,44,45,46,47]. In addition, it provides a deeper explanation of stress-related situations and the outcomes by examining the link between a person and their situation, which is underpinned by psychological strains and the changes an individual adopts to their behavioral outcomes in order to avoid potentially detrimental consequences these strains may cause [43].

#### 2.1.1. Stressor (Social Media Overloads)

With the growing number of social media platforms, users will be overloaded and overwhelmed by the amount of time spent online. In the context of this study, this undesirable condition is characterized as WhatsApp overload. Scholars have theorized and empirically validated the dimensions of social media overloads in a different study context. However, studies on social media as determinants of social media overload in the workplace remain scarce [19,27]. The potential work-related consequences of social media stressors, especially in the context of innovative job performance, have remained understudied. In addition, most prior studies on social media overload have been concentrated more on generic social media users [28,29,30] and students [31,32] compared to employees at work. Furthermore, studies on WhatsApp as the primary social media use are scarce, presenting a significant gap in the literature.

This study considers social media overload a significant techno-stressor that could negatively impact employees’ job performance [7,19]. This suggested that WhatsApp overload is notably a substantial source of workplace stress due to its widespread usage. WhatsApp use at work can increase the risk of suffering from psychological consequences caused by social media overloads. Furthermore, studies on the various dimensions of social media overload, including communication overload, information overload, system, and social overload, are scarce [30]. The interconnection of the different social media overloads on employees’ job performance remains limited. Therefore, this study offers a significant extension of social media overload as a consequence of WhatsApp usage consisting of communication overload, information overload, and social overload as representative stressors in understanding the psychological mechanism underlying technostress on employees’ innovative job performance.

#### 2.1.2. Strain (Technostress)

Technostress is considered a modern disease that results from an individual inability to cope or adapt with new technologies in a healthy manner. In addition, technostress can be explained as a stress response that negatively impaired user’s physical, psychological, or behavioral state as a result of excessive technology usage [48]. The most common psychological consequences of technostress are burnout, anxiety, depression, and perceived social pressure, all of which have a detrimental impact on an individual’s productivity [49]. Furthermore, individuals who experience technostress physically tend to feel light-headed, sweat, or experience heavy breathing [50]. Thus, this study will discuss the effect of social media overloads on employees’ technostress predicted by WhatsApp use at work.

#### 2.1.3. Outcome (Innovative Job Performance)

In today’s digital era, social media is recognized as a form of computer-mediated communication that strongly influences employees’ job performance. However, several studies have shown inconsistent findings with job performance, as existing literature has discovered different results on the impact of social media use on job performance. For instance, recent studies by [1,3,51] discovered that social media usage positively increased employees’ job performance. In contrast, empirically, social media use at work was associated with adverse job performance [19,23]. Despite the prevalence of prior studies on social media use at work, empirical evidence suggests that social media usage does not influence employees’ job performance [14]. Due to the inconsistent findings, this study attempts to fill in the gap in the body of knowledge on employees’ use of WhatsApp at work. The findings would explain the dual effect of WhatsApp usage on employees’ innovative job performance

### 2.2. Hypothesis Development

#### 2.2.1. Social Media Use and Information Overload

Employees can solve problems and make decisions by searching for information through social media for work-related purposes. It provides accessibility to massive amounts of information and expands their mental capability [52]. In consequence, they will spend more time and effort in screening excessive information in order to obtain useful information related to job purposes [19]. Some scholars have discovered the association between social media use and information overload. For instance, [27] found that Korean office workers experienced information overload from social media use for work-related purposes, specifically mobile instant messaging services (MIMs) during working hours, which significantly increased employee burnout and turnover intention. As social media platforms on smartphones provide instant access to information around the clock, users are inundated with irrelevant information about their personal lives, events, group conversations, brand-related promotions and news [53]. Undoubtedly, social media use at work may result in information overload, as employees are exposed to overwhelming volumes of information that exceed their’ cognitive capabilities to process the information. Therefore, the first hypothesis is presented as:

**H1:** 
*Social media use at work positively influences employees’ information overload.*


#### 2.2.2. Social Media Use and Communication Overload

The existence of multiple communication channels from various social media platforms may lead to frequent communication from different users, i.e., colleagues, family, and friends. Constant communication via social media may distract their attention and cause excessive interruptions in their work activities, resulting in low productivity [15,16]. As a consequence, it causes communication overload, which increases the cognitive burden on the user and leads to interruption, consequently impeding work tasks [54]. Previous studies empirically discovered the association between social media use and communication overload among employees. For instance, [19] found that employees in China suffered from communication overload, significantly predicted by social media use in organizations, which interfered with their work performance. Employees who experienced communication overload as a result of social media use for work-related purposes are more likely to increase burnout and employee turnover intention [27]. In addition, employees may find it difficult to refocus their attention since handling online communications and other tasks requires significant cognitive effort [16]. Therefore, the second hypothesis is proposed:

**H2:** 
*Social media use at work positively influences employees’ communication overload.*


#### 2.2.3. Social Media Use and Social Overload

Pervasiveness social media usage has integrated into employees’ social life, socialization, and networking via social media within the organization [4,55]. During work, employees encounter social overload when they continuously receive communication from colleagues, friends, and family. This intensive social interaction leads to social burden on users [16]. This phenomenon requires users to invest effort and time in responding to social requests and interactions. The authors of [19] reported that employees in China who excessively engage in social media suffered from social overload. The authors of [35] empirically addressed social overload as a dark side of technology associated with people’s lives in social media. Thus, this study proposed that social media use at work influences social overload among employees. The third hypothesis is as follows:

**H3:** 
*Social media use at work positively influences employees’ social overload.*


#### 2.2.4. Information Overload and Technostress

Employees who experience information overload are more likely to suffer from technostress when they receive a high volume of information from different social media platforms that exceed their cognitive limit for information processing. Several studies have discovered that information overload is a significant problem related to technostress in the workplace. Past research shows that employees who were excessively occupied with social media at work experienced information overload, resulting in technostress in the form of social media exhaustion [19]. Furthermore, [53] revealed that employees in South Asia experienced social media fatigue in the workplace due to information overload. Indeed, employees need to increase their information processing capabilities to sift through and organize overwhelming information from social media, which may lead to technostress. The following hypothesis is deduced as:

**H4:** 
*Information overload positively influences employees’ technostress.*


#### 2.2.5. Communication Overload and Technostress

Communication overload using social media has been identified as a stressor that negatively impacts users’ psychological states [32,53,54]. Previous studies have empirically discovered that communication overload induced technostress among employees in the work setting [56,57]. Employees bear communication overload when they received excessive demands for communication through social media and were positively associated with social media exhaustion [19]. Individuals encountering excessive communication from coworkers, family, and friends face communication overload, which results in loss of concentration and interrupting work tasks. Subsequently, it leads to technostress due to exhaustion and overwhelming feelings as they cannot handle the situation effectively. Hence, the next hypothesis is denoted as:

**H5:** 
*Communication overload positively influences employees’ technostress.*


#### 2.2.6. Social Overload and Technostress

The pervasiveness of social media has integrated employees’ social life, as they have a wide range of connections in social media, including socialization and networking within the organization [37]. In this situation, users respond to the request for social support as a duty to fulfill their role in giving support in order to maintain a social media relationship [30]. Employees who use social media at work are more likely to experience excessive social demands from other social media users, including coworkers, friends, and family, which increases the risk of suffering from social overload if they are unable to adequately address the problem effectively. When the social requests exceed employees’ emotional and cognitive capacity, they may feel stressed, subsequently affecting their work tasks. In this study, social overload is regarded as one of the stressors influencing employees’ technostress. Therefore, the following hypothesis is presented as:

**H6:** 
*Social overload positively influences employees’ technostress.*


#### 2.2.7. Technostress and Innovative Job Performance

Social media overloads have emerged as part of a stressor that cause users’ technostress in the form of exhaustion fatigue. Employees who suffer from overload often feel tired to perform their work task due to information, communication, and social demand, exceeding their cognitive capacities [27,53]. Consequently, they will increase their time, energy, and emotional resources on work tasks. In this regard, exhausted employees do not have sufficient resources to complete the required work task, resulting in a decline in their job performance. Furthermore, social media overloads were found to induce technostress among employees, negatively impacting their job performance [7,19,58,59].

Moreover, technostress has also negatively affected employees’ innovativeness due to the imbalanced relationship between environmental demands (social media use) and exceeding their coping abilities. This threat can negatively impair their creative and critical thinking to produce or adopt, promote, and implement novel ideas [50,60,61]. Therefore, it is posited that technostress resulting from social media overloads may reduce innovative performance, as supported by previous studies [60,62]. Therefore, the hypothesis is as follows:

**H7:** 
*Technostress has a negative influence on employees’ innovative job performance.*


Figure 1 presents the conceptual framework of this study.

## 3. Research Methodology

### 3.1. Sampling

Utilizing a quantitative approach, a survey based on a self-administered questionnaire was administered. The samples are composed of Malaysian public and private employees. Since public sector employees are the backbone of the country in providing outstanding public services, implementing measures at the individual level may enhance an organization’s overall performance [63,64]. In addition, the government’s success largely depends on the employees’ ability, high cognitive skills, and work performance in demonstrating their knowledge-based service. However, despite its significance, research on public sector employees’ job performance has received little attention [65], especially on innovative performance. In addition, employees from the private sector, specifically GLCs in which Government holds a certain amount of shares, also play a major role in contributing to the economic growth of Malaysia as money, returns, and profits have always been the key elements in the private sector [66]. Thus, this study focuses on employees from the public and private sectors.

A quota sampling frame was applied to ensure sample representation within the population. The employees were divided into government agencies, ministries, statutory bodies, and the private sector (government-linked companies, GLCs) based on a 40:40:20 ratio, respectively. This study used the G*Power application to draw the sample size from the target population. Further, with a power of 0.8 and an effect size of f^2^ = 0.15, in addition to the predictor of the variable with the highest value of 3, the minimum sample was determined as 77.

### 3.2. Inclusion and Exclusion Criteria

In this study, there are three inclusion criteria for the target population: (1) government and private sector employees, (2) utilized social media for work purposes, and (3) used only WhatsApp, Facebook, Twitter, LinkedIn, or Telegram. Firstly, employees from government and private (government-linked companies, GLCs) sectors are included in this study. Next, employees who actively use social media to manage work tasks such as communication, offer knowledge-based service, gather essential evidence/information, or professional networking. Lastly, with various social media platforms available for the public, this study only focuses on the employees who utilized WhatsApp, Facebook, Twitter, LinkedIn, or Telegram for work and professional purposes. Meanwhile, the exclusion criteria for the target population are job level and use of the organizational account. As this study examines the disadvantages and advantages of social media use at work on innovative job performance, the position level is excluded due to the nature of the job related to knowledge-based service. Employees who use organizational accounts are excluded from the target population because they may cause an invasion of the organization’s privacy and lead to conflictual situations in the workplace. Table 1 sets out the inclusion and exclusion criteria of the sampling.

### 3.3. Data Collection Procedure

An online survey was utilized for data collection. The target population was employees who use social media for work-related purposes. The questionnaires were distributed through email to 1500 respondents. Respondents filled in a Google form attached in the email. To increase response rate, follow-up email was sent to the participants. To this end, 208 responses were obtained within three months (November 2021 to January 2022), with a response rate of 13.8%.

### 3.4. Measurement

The measures of all constructs in this study were adapted from previously validated scales and items. First, the three items for social media use at work were adapted from [12]. Second, information overload was measured using four items adapted from [67]. Third, communication overload adapted the five items from [16] and social overload was measured using five items adapted from [19]. Next, technostress was measured using five items adapted from [68]. Lastly, the measurement for innovative job performance was adapted from [11]. All the items were rated on a 7-point Likert scale. In summary, the list of measurement items for the constructs is presented in Appendix A. Meanwhile, the list of variables included with the number of items assessed in the present study is depicted in Table 2.

### 3.5. Data Analysis

The data analysis began with descriptive statistics and this study utilized SPSS version 22 (IBM Corp., New York, NY, USA) to measure the frequency of background characteristics. In addition, this study applied the PLS-SEM approach in analyzing the actual data using [69] SmartPLS 3.0 software. The PLS-SEM method helps estimate complex structural models with many constructs, indicators, and/or model relationships, as well as its ability to adequately use non-normal data [70]. In addition, it is able to perform exploratory research in developing theory and estimate model that commonly displays a high degree of statistical power compared to the CB-SEM [71,72,73]. In addition, advanced model elements, such as hierarchical component models, moderator variables, or nonlinear relationships, can be handled flexibly using PLS-SEM [74,75,76]. Thus, in exploring the association of social media and innovative job performance in this study, PLS-SEM is deemed appropriate.

## 4. Results

### 4.1. Data Preparation

Data screening and cleaning will result in data exclusion. The reasons for data to be excluded are because of the straight lining, missing values, and redundant responses from the same respondents. The first step in data cleaning is checking the blank response among the collected responses. To treat the blank responses, this study used the formula COUNTBLANK (item1 to item28) in Microsoft Excel (Microsoft Corporation, Washington, DC, USA) before transferring the data into SPSS. The results found no blank response in the data, confirming the respondents answered all questions. Thus, no missing value data issues were reported in this study since the questions in the Google form were arranged in mandatory responses. The data were automatically stored in the Google form. Hence, treatment for missing values is not implemented in this study.

The authors of [77] explained that straight lining occurs when respondents provide identical answers to the questions using the same response scale. This study treated the straight-lining assessment by applying the formula standard deviation function in Microsoft Excel 2016. Using the formula of STDEV (item1 to item28), 2 of 208 responses were detected with straight lining with zero values. Therefore, these two responses were excluded from further analysis, leaving 206 remaining responses to be analyzed.

### 4.2. Demographic Information

Table 3 presents the demographic information of the 206 respondents. With reference to gender, 46.1 percent (95) were male, whereas 53.9 percent (111) were female. In terms of education, the majority of respondents (57.8 percent) are Bachelor’s degree holders and Master’s degree holders (23.3 percent). Furthermore, 28.2 percent of respondents have worked for more than 11–15 years and 6–10 years (19.9 percent). The social media platforms frequently used are WhatsApp (88.3 percent) and Facebook (5.8 percent). As expected, Malaysian employees mostly use WhatsApp as a medium to communicate and interact for work-related purposes due to the compliance, internalization, and identification that influenced the employees to use WhatsApp in their routine work. In addition, The Digital News Report (2017) found that Malaysians are the world’s largest users of WhatsApp at 51%. This result suggests that the predominantly utilized social media platform by Malaysian employees in the workplace is WhatsApp.

### 4.3. Common Method Bias

A statistical remedy was employed in this study to manage common method bias, which is common in behavioral research [78,79]. As such, [80] suggested that in PLS-SEM, a full collinearity test can be used to assess common method bias and a variance inflation value (VIF) below 3.3 indicates the dataset does not suffer common method bias. There is no significant issue in the dataset, as the VIF values of all constructs are lower than 3.3, as shown in Table 4.

### 4.4. Measurement Model

The measurement model is the first stage of using PLS-SEM that specifies the relations between the latent variable (construct) and its indicator (manifest variable). The purpose of measurement model analysis is to ensure all the required relationships between the latent variables and the indicator are met by the model assessment [75]. For construct reliability and validity, the convergent validity is evaluated by assessing the factor loadings and average variance extracted (AVE). Cronbach’s alpha and composite reliability are for the internal consistency reliability [74,81]. Table 5 shows that all the factor loadings exceed the minimum of required value 0.6 for an exploratory study [82]. Meanwhile, the Cronbach’s alpha and composite reliability for all constructs were higher than the required value of 0.7 [82]. Table 5 presents the measurement model’s construct validity and reliability.

Discriminant validity is essential to ensure that each variable is distinct and not related to each other [74]. This study applied the Heterotrait–monotrait (HTMT) ratio of correlation, which has better performance in measuring the discriminant validity in variance-based SEM compared to the cross-loadings and Fornell Larcker criterion [83]. The authors of [84] stated that a cut-off value of HTMT for conceptually dissimilar constructs is less than 0.85, while conceptually similar constructs are less than 0.9, establishing the discriminant validity that reliably distinguishes between those pairs of latent variables, depending on the study context. Table 6 shows that all the correlation values are below 0.85, suggesting that the variables in this study possess satisfactory discriminant validity.

### 4.5. Structural Model

This study implemented Mardia’s multivariate kurtosis to measure the normality of data. As [85] suggested, this study uses the online tool to calculate univariate/multivariate skewness and kurtosis at http://webpower.psychstat.org/models/kurtosis/ (accessed on 2 August 2022). The results indicate that data were not multivariate normal, as shown by the skewness (*β* = 8.351, *p* < 0.01) and kurtosis (*β* = 62.962, *p* < 0.01). This calls for using a nonparametric analysis tool, SmartPLS 3.2.8, to perform bootstrapping.

Following the suggestion of [70], a 5000 bootstrapping re-sampling technique was performed to assess the structural model based on the path coefficient and statistical significance [86]. To test the model with different research hypotheses, the path coefficient of exogenous to endogenous variables by the *β*-value, *t*-values, and squared multiple correlation (R^2^) values of explained variance on the endogenous variable were evaluated. Table 7 shows the result of R^2^, f^2^, and Q^2^; meanwhile, Table 8 displays the structural analysis results and decision on hypotheses, while Figure 2 illustrates the structural path.

The result shows that five of the seven proposed hypotheses were supported. As hypothesized, H1, H2, and H3 were supported as social media use at work has a positive influence on information overload (*β* = 0.234, *t* = 3.078), communication overload (*β* = 0.216, *t* = 2.891), and social overload (*β* = 0.164, *t* = 2.276). The R^2^ of the three variables are 0.055 (Q^2^ = 0.031), 0.047 (Q^2^ = 0.031), and 0.027 (Q^2^ = 0.017), denoting that social media use at work explained 5.5%, 4.7%, and 2.7% of the variance, respectively. Next, H4 and H5 were accepted, which posited that information overload (*β* = 0.342, *t* = 3.001) and communication overload (*β* = 0.343, *t* = 3.064) showed a significant positive effect on employees’ technostress; meanwhile, social overload on employees’ technostress was not significant (*β* = −0.089, *t* = 1.045), thus, rejecting H6. The R^2^ was 0.343 (Q^2^ = 0.270), indicating that the three overloads explained 34.3% of the variance on technostress. Lastly, H7 (*β* = 0.065, *t* = 0.506) demonstrated that technostress was found to be insignificant towards innovative job performance among employees, with an R^2^ of 0.004 (Q^2^ = 0.000), which indicates that technostress explained 0.4% of the variance in innovative performance.

## 5. Discussion

The study utilized the SSO model to examine how social media use at work predicts social media stressors (information overload, communication overload, and social overload), strain (technostress), and, subsequently, relationship on their innovative job performance. First, the study presented the new outcomes since WhatsApp is a predominant social media platform that predicts social media overloads in the Malaysian workplace. The finding discovered its influence on information overload, communication, and social overload. Based on the findings, H1 (*t* = 3.078), H2 *(t* = 2.891), and H3 (*t* = 2.276) showed a *t*-value of more than 1.65 and these hypotheses were supported as WhatsApp use at work positively influences information overload, communication overload, and social overload. Even though the findings revealed that the impact of WhatsApp usage is very mild on the three stressors, nevertheless, it significantly stimulates social-media-related overload statistically, hence, contributing to users’ technostress. Consistent with prior studies [16,19], pervasive social media access led to technology overload. Due to the integration of social media into work life, employees experience challenges in balancing their professional and private life because work responsibilities may be accessible anywhere and at any time [87]. With the wide range of social features in social media, a large volume of information, communication, and social interaction may be generated for personal and professional reasons, necessitating employees to process and attend to, indirectly leads to social media overload.

Additionally, H4 (*t* = 3.001) and H5 (*t* = 2.891) were accepted as a t-value of more than 1.65, positing that information overload and communication overload significantly positively affected employees’ technostress. The findings of this study show that information overload and communication overload contributed to the pervasive phenomenon of technostress among Malaysian employees. As anticipated, these findings are in line with previous studies and highlight the adverse consequence of information overload and communication overload on individuals’ psychological well-being [18,30,54]. Employees who experience information and communication overload are unable to keep up with the high volume of information and communication generated by WhatsApp, as it exceeds their cognitive abilities to process it. This undesirable condition would influence employees’ execution and decision-making capability, resulting in technostress [88].

Meanwhile, social overload on employees’ technostress was not significant as *t* = 1.045, *t*-value < 1.65, thus, rejecting H6. This study found that social overload did not contribute to employees’ technostress. Therefore, this finding is inconsistent with past studies [30,35,67]. Social overload mainly focuses on private activities that can be ignored temporarily during work hours and processed later in the desired sequence, contrary to information overload and communication overload, which must be handled immediately at work due to the association with work tasks [19]. In addition, H7 (*t* = 0.506) was rejected, indicating a *t*-value < 1.65, demonstrating that technostress was found insignificant towards innovative job performance among Malaysian employees. This result suggests that employees showed effective coping strategies for minimizing, tolerating, and coping with technostress related to exceptional work or social demands from WhatsApp at the workplace. With good coping abilities, this strain and technostress had no effect on employees’ innovative job performance. Hence, employees can show their creative and critical thinking by producing, adopting, promoting, and implementing novel ideas.

## 6. Conclusions

### 6.1. Theoretical and Practical Implications

By employing the SSO model, this study contributes to the theoretical understanding of the role of social media use among employees in government departments. The present study provides an essential extension of current research in the form of a detailed theoretical understanding of the psychological mechanism underlying employees’ work behavior, specifically in innovative job performance. In addition, this study examines the process of how WhatsApp use at work plays a significant role in employee’s innovative job performance from the perspective of SSO. The outcome provides a meaningful theoretical contribution to the literature on work-related stress and the effects of WhatsApp use on employee outcomes at the workplace.

It is possible that employees who use WhatsApp at work may be oblivious to the potential fallout from their social media usage. They view WhatsApp as an integral part of their daily life. Considering the fact that Malaysia constitutes the highest number of internet users in the Southeast Asian region, employees should have a greater insight and understanding of WhatsApp usage and its dual consequences (positive or negative) at the workplace. Employees may implement measures to regulate their habits on WhatsApp use in order to avoid adverse consequences on their work behavior, particularly innovative job performance. In addition, the findings of this study offer management with guidance to adopt an emerging and popular technology, social media, especially WhatsApp, as a medium to foster innovativeness among employees. Such a measure can effectively provide practical insights for management to create new strategies in mitigating issues of personal social media usage at work. In addition, management is able to reinforce existing guidelines or policies surrounding the usage of WhatsApp at work to promote better routine and innovative performance, thus, benefiting organizations and ensuring psychological and mental health for a work–life balance among employees.

### 6.2. Limitations and Suggestions for Future Work

Although this study offers valuable insights, certain limitations should also be acknowledged and addressed for future studies. First, given the pervasiveness of social media overload and its effects on employees’ innovative job performance, our findings imply that future research should pay closer attention to social media overload, its antecedents, and its outcomes in other types of organizational settings (i.e., private, large corporate organization) or specific industries, such as manufacturing, services, and education. Different contexts would provide a detailed and different understanding of adapting to social media usage, which might influence employees’ innovative job performance.

Second, this study highlighted solely on the negative consequences of social media use at work. Future studies should explore both the harmful and beneficial effects of social media use at work by integrating two theories/models (e.g., the SSO model and social capital theory). We believe that researchers in other disciplines can improve the understanding of social media use at work and employees’ outcomes by combining theories from different, multiple perspectives.

Third, this study performed a cross-sectional design, in which data were collected from a single source response. Although the statistical result revealed no evidence of common method bias, it is possible that respondents might be unable to provide accurate information when responding to sensitive questions about their physical and mental health. Future scholars should consider applying a mixed-method design by adding interview sessions or observation to accurately measure employees’ technostress and innovative job performance.

Next, this study was carried out with users who predominantly utilized WhatsApp as their medium for work-related purposes. Hence, the findings might be restrained and impractical to other social media platforms. Furthermore, the generalizability of the current study is somewhat limited because the interaction on WhatsApp is often limited to close contacts or specific social groups and not intended for the public at large [20]. Therefore, scholars should consider replicating the study method in different social media platforms at work, either for combined or exclusive usage. Then, the different platforms would produce an exact effect of social media usage at work on social media overloads.

This study also has geographical limitations that prevent generalization to other countries. Since this study was performed in Malaysia, the findings only apply to users with comparable demographic information and similar work culture. Thus, scholars should use this model as a foundation for future studies on similar respondents from different countries to strengthen the results’ validity and reliability.

Lastly, future scholars should consider conducting a cross-sectional or longitudinal study by reversing causation in the relationship between social media and stress, especially in the workplace setting. This study focused on the roles of social media toward technostress issues by discussing the consequences of social media use at work. To gain a clear picture of the social-media–technostress link, the direction of social media causing psychological well-being problems might instead be reversed. As [89] mentioned, future research needs to move toward a deeper analysis of the reverse possibility underlying social media use.

Overall, this study presents the exact mechanism of crucial roles of WhatsApp use at work and its influence on employees’ innovative performance. The discussion of the findings shows that most of the hypotheses are congruent with prior studies. This study suggests that WhatsApp use at work mildly predicts social media overloads, including information overload, communication overload, and social overload. Furthermore, the findings found that these overloads, except social overload, induced employees’ technostress. However, this strain indicates that technostress has not significantly affected employees’ innovative performance. The findings of this study provide an essential extension of prior knowledge on the conceptual relationships for social media overloads that were empirically validated in terms of technostress and employee outcomes. Moreover, the outcomes of this study will be of immense benefit to both employees and employers in enhancing the association between social media use at work and increasing employees’ innovative job performance.

## Figures and Tables

**Figure 1 behavsci-12-00456-f001:**
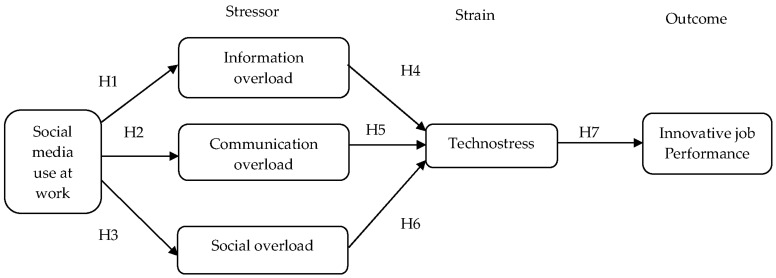
Proposed conceptual framework.

**Figure 2 behavsci-12-00456-f002:**
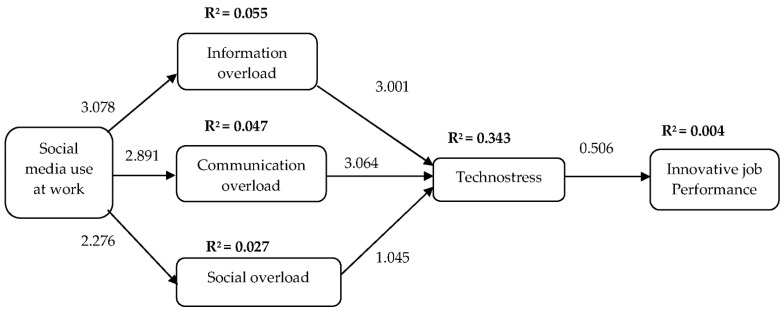
Structural Model.

**Table 1 behavsci-12-00456-t001:** Inclusion and exclusion criteria.

	Inclusion Criteria	Exclusion Criteria
1	Government and private employees	Job position
2	Actively used social media for work purposes.	Use an organizational account.
3	Use WhatsApp, Facebook, Twitter, LinkedIn, or Telegram.	

**Table 2 behavsci-12-00456-t002:** The list of variables measured in this study.

Constructs	Number of Items	Source Adaption	Unit Analysis
Social media use at work	3	[12]	Professional employees were working in higher education from China.
Information overload	4	[67]	Social media users who used Qzone from China.
Communication overload	5	[19]	Students in a Chinese university.
Social overload	5	[19]	Chinese employees from various industries (e.g., education, manufacturing, services etc.).
Technostress	5	[68]	Frontline service employees.
Innovative job performance	6	[11]	Employees of multinational Information Technology company.
Total	28		

**Table 3 behavsci-12-00456-t003:** Demographic Information.

Categories	Type	Frequency (*n*)	Percentage (%)
Gender	Male	95	46.1
	Female	111	53.9
Age	25–30	29	14.1
	31–35	38	18.4
	36–40	48	23.3
	41–45	45	21.8
	46–50	18	8.7
	51–55	9	4.4
	56–60	18	8.7
	Above 60	1	0.5
Level of education	SPM/A-level/Certificate	5	2.4
	STPM	2	1.0
	Diploma	28	13.6
	Degree	119	57.8
	Master	48	23.3
	PhD	4	1.9
Years of working	5 years and below	32	15.5
	6–10 years	41	19.9
	11–15 years	58	28.2
	16–20 years	36	17.5
	21–25 years	14	6.8
	26–30 years	8	3.9
	More than 30 years	17	8.3
Sector	Government, ministries	82	40.0
	Government, statutory bodies	82	40.0
	Private, GLC	42	20.0
Social media platforms	WhatsApp	182	88.3
	Telegram	2	1.0
	Facebook	12	5.8
	Twitter	1	0.5
	Instagram	3	1.5
	Others	6	2.9

**Table 4 behavsci-12-00456-t004:** Full Collinearity.

Construct	Social Media Use at Work	Information Overload	Communication Overload	Social Overload	Technostress	Innovative Job Performance
VIF	1.123	2.414	2.911	1.765	1.571	1.038

VIF = variance inflation factor.

**Table 5 behavsci-12-00456-t005:** Reliability and validity analysis.

Construct	Items	Loadings	Cronbach	Composite Reliability (CR)	Average Variance Extracted (AVE)
Social media use at work	SM 1	0.857	0.871	0.921	0.796
	SM 2	0.911			
	SM 3	0.908			
Information Overload	IO1	0.899	0.848	0.899	0.691
	IO2	0.896			
	IO3	0.813			
	IO4	0.702			
Communication Overload	CO1	0.843	0.941	0.936	0.744
	CO2	0.837			
	CO3	0.834			
	CO4	0.909			
	CO5	0.886			
Social Overload	SO1	0.911	0.941	0.955	0.810
	SO2	0.896			
	SO3	0.894			
	SO4	0.913			
	SO5	0.886			
Technostress	TECH1	0.863	0.961	0.970	0.866
	TECH2	0.954			
	TECH3	0.954			
	TECH4	0.935			
	TECH5	0.941			
Innovative job performance	IP1	0.825	0.941	0.945	0.743
	IP2	0.803			
	IP3	0.804			
	IP4	0.921			
	IP5	0.927			
	IP6	0.883			

**Table 6 behavsci-12-00456-t006:** Heterotrait–Monotrait ratio of correlations (HTMT).

Construct	Communication Overload	Innovative Job Performance	Information Overload	Social Media Use at Work	Social Overload	Technostress
Communication Overload						
Innovative job performance	0.094					
Information Overload	0.830	0.082				
Social media use at work	0.242	0.195	0.257			
Social overload	0.698	0.086	0.601	0.176		
Technostress	0.559	0.058	0.606	0.048	0.328	

**Table 7 behavsci-12-00456-t007:** The result of the coefficient determination (R Square), effect size (f Square), and predictive relevance (Q Square).

Hypothesis	Relationship	R^2^	f^2^	Q^2^
H1	Social media use at work → Information overload	0.055	0.058	0.031
H2	Social media use at work → Communication overload	0.047	0.049	0.031
H3	Social media use at work → Social overload	0.027	0.028	0.017
H4	Information overload → Technostress		0.082	
H5	Communication overload → Technostress		0.066	
H6	Social overload → Technostress	0.343	0.007	0.270
H7	Technostress → innovative job performance	0.004	0.004	0.000

**Table 8 behavsci-12-00456-t008:** Hypothesis testing.

Hypothesis	Relationship	Path Coefficient (*β*)	Std Dev	*t*-Value	*p*-Value	BCI LL	UCI LL	Decision
H1	Social media use at work → Information overload	0.234	0.076	3.077	0.001	0.109	0.353	Accepted
H2	Social media use at work → Communication overload	0.216	0.076	2.846	0.002	0.089	0.336	Accepted
H3	Social media use at work → Social overload	0.164	0.074	2.212	0.014	0.036	0.278	Accepted
H4	Information overload → Technostress	0.342	0.114	2.999	0.001	0.128	0.506	Accepted
H5	Communication overload → Technostress	0.343	0.112	3.066	0.001	0.158	0.526	Accepted
H6	Social overload → Technostress	−0.089	0.087	1.025	0.153	−0.228	0.056	Rejected
H7	Technostress → innovative job performance	0.065	0.13	0.505	0.307	−0.242	0.155	Rejected

BCI LL = Bias confidence interval lower limit, BCI UL = Bias confidence interval upper limit.

## Data Availability

Not applicable.

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
