# Peer review of "The Effect of WhatsApp Usage on Employee Innovative Performance at the Workplace: Perspective from the Stressor–Strain–Outcome Model"

_behavsci, 2022, doi:10.3390/bs12110456_

Round 1

Reviewer 1 Report

I would like to commend the authors for providing a clearly written and logically structured manuscript. However, I have some comments that need to be addressed as per the content of the paper:

1.) The responses (206) are a small fraction of the sample of participats (1500 respondents). The authors should provide an analysis of why the sample of participants is representative of the original sample. Otherwise, there is room to assume that a certain category of people may have provided their responses (e.g. only the younger responses, or any other group).

2.) The authors state that "The measures of all constructs in this study were adapted from previously validated scale and items", and go on to explain the sources for each section of the questionnaire. However, there is no indication of what changes were made to the original questions. Moreover there are no sample items for each scale, so that the reader may understand how each construct was measured. Finally, the scales were not provided (Likert scales with what kind of text field for each number value 1-7 for each scale?). All this would be resolved, if the authors provide the questionnaire along with the measurement scales for each construct, in an Appendix or table.

3.) In Table 1 , we see that SM platforms usage is mainly in WhatsApp and not any of the rest of the platforms. However, WhatsApp is more of a messaging service than a social media service. Hence, if the participants in the study mainly used WhatsApp, the hypothesis that social media usage is at play here doesn't look right. Perhaps "Social messaging" usage may be more appropriate for the authors' hypotheses. If I am mistaken in interpreting the table, the table may need clarification itself. It looks like this study is more on the effect of using WhatApp (a messaging service), than social media usage. If that is so, it may be appropriate to re-position the paper accordingly, changing also its title.

4.) The explanation of the statistical results look a bit strange to me (see page 10 for example). In my understanding, R-squared values range from 0 to 1 and are commonly stated as percentages from 0% to 100%. An R-squared of 100% means that all movements of a security (or another dependent variable) are completely explained by movements in the index (or the independent variable(s) you are interested in). Therefore, I would expect that R2 = 0.343 over technostress in the figure would mean that Information overload, communication overload, and social overload, jointly explain 34.3% of the variance in technostress. However, the authors explain this very differently both in numbers, as well as in essence in the paragraph before Figure 3. The same stands for all other numbers in the figure. Therefore, if my view is correct, the results are not explained correctly by the authors. Moreover, it may seem that SM use at work only explains a small percent of the social overload of employees (2.7%), communication overload (4.7%), and information overload (5.5%). Moreover, technostress seems to explain a fraction of the variance in innovative performance (0.4%). Therefore, what I make of the results is that the only strong relationship validated is that Info/Comms/Social overload jointly explain a significant percentage in the variance in technostress. Please review your results according to the above and adjust as needed their discussion and your conclusions.

Author Response

Reviewer 1

No

Comments

Responds

1.        

The responses (206) are a small fraction of the sample of participats (1500 respondents). The authors should provide an analysis of why the sample of participants is representative of the original sample. Otherwise, there is room to assume that a certain category of people may have provided their responses (e.g. only the younger responses, or any other group).

Thank you, reviewer 1

We have provided a better explanation regarding the participants in 3.1 as suggested by reviewer 1.

2.        

The authors state that "The measures of all constructs in this study were adapted from previously validated scale and items", and go on to explain the sources for each section of the questionnaire. However, there is no indication of what changes were made to the original questions. Moreover there are no sample items for each scale, so that the reader may understand how each construct was measured. Finally, the scales were not provided (Likert scales with what kind of text field for each number value 1-7 for each scale?). All this would be resolved, if the authors provide the questionnaire along with the measurement scales for each construct, in an Appendix or table.

Thank you, reviewer 1

We have provided the list of items (original and modified) for each construct in Appendix A.

3.        

In Table 1 , we see that SM platforms usage is mainly in WhatsApp and not any of the rest of the platforms. However, WhatsApp is more of a messaging service than a social media service. Hence, if the participants in the study mainly used WhatsApp, the hypothesis that social media usage is at play here doesn't look right. Perhaps "Social messaging" usage may be more appropriate for the authors' hypotheses. If I am mistaken in interpreting the table, the table may need clarification itself. It looks like this study is more on the effect of using WhatApp (a messaging service), than social media usage. If that is so, it may be appropriate to re-position the paper accordingly, changing also its title.

Thank you, reviewer 1

Today, WhatsApp allows users to have a profile, timeline posts (WhatsApp stories), share statuses, and freely exchange video, audio, text, and images similar to Facebook, Twitter, Instagram, and YouTube. Besides,

WhatsApp also gives users the option of having a Mobile App, Web Version, or Desktop App. These show that WhatsApp is a social media.

We believe WhatsApp is also part of social media according to a definition by Obar and Wildman (2015), the definition of social media presented in the literature and the following commonalities among current social media services are: 1) Social media services are currently (currently) Web 2.0 Internet-based application, 2) User generated content is the lifeblood  of social media, 3) Individual and group creator user-specific profile for a sit or app designed and maintained by a social media services 4) Social media services facilitate the development of  social network online by connecting a profile with those of other individuals or/and groups.

4.        

The explanation of the statistical results look a bit strange to me (see page 10 for example). In my understanding, R-squared values range from 0 to 1 and are commonly stated as percentages from 0% to 100%. An R-squared of 100% means that all movements of a security (or another dependent variable) are completely explained by movements in the index (or the independent variable(s) you are interested in). Therefore, I would expect that R2 = 0.343 over technostress in the figure would mean that Information overload, communication overload, and social overload, jointly explain 34.3% of the variance in technostress. However, the authors explain this very differently both in numbers, as well as in essence in the paragraph before Figure 3. The same stands for all other numbers in the figure. Therefore, if my view is correct, the results are not explained correctly by the authors. Moreover, it may seem that SM use at work only explains a small percent of the social overload of employees (2.7%), communication overload (4.7%), and information overload (5.5%). Moreover, technostress seems to explain a fraction of the variance in innovative performance (0.4%). Therefore, what I make of the results is that the only strong relationship validated is that Info/Comms/Social overload jointly explain a significant percentage in the variance in technostress. Please review your results according to the above and adjust as needed their discussion and your conclusions.

Thank you, reviewer 1

We have addressed all the comments and amended accordingly.

Reviewer 2 Report

This is an interesting study examining the role of social media usage as a predictor of employee performance. The paper is well-written and I believe it may contribute to the literature well. I only have several comments to further improve the manuscript further:

1. First, I would suggest the authors to avoid abbreviate social media as SM. It will help to improve the readability of the manuscript.

2. There is a need for a stronger conceptualization of what is social media in the Introduction. Given that many of the participants use Whatsapp (88%) as their social media, it may not be the same social media like Facebook or Twitters. Perhaps, more justification is necessary regarding what is constituted social media in the current study and how it may lead to stress and overload. 

3. It will be important for the authors to elaborate further how the quota sampling was conducted. Also, there is a need to supplement more information regarding the inclusion and exclusion criteria of the sampling

4. There is a need to mention how many missing data and how these missing data were treated in the current study. Relevant paper:
Newman, D. A. (2014). Missing data: Five practical guidelines. Organizational Research Methods, 17(4), 372-411.

5. One limitation that I hope the authors can elaborate and discuss in their discussion is the possibility of reverse causation in the relationship between social media and stress. Given the use of cross-sectional design, it is still possible that stress is not a consequence but antecedent of social media use. This issue should be discussed and acknowledged and will help to provide a balanced perspective in this topic. See the following paper that will be relevant and helpful to the discussion: Does social media use increase depressive symptoms? A reverse causation perspective. (2021). Frontiers in Psychiatry, 12, 641934.

Author Response

Reviewer 2

No

Comments

Responds

1.        

 First, I would suggest the authors to avoid abbreviate social media as SM. It will help to improve the readability of the manuscript.

2.        

There is a need for a stronger conceptualization of what is social media in the Introduction. Given that many of the participants use Whatsapp (88%) as their social media, it may not be the same social media like Facebook or Twitters. Perhaps, more justification is necessary regarding what is constituted social media in the current study and how it may lead to stress and overload. 

Thank you, reviewer 2

We have provided a better explanation in 1.0 Introduction (first paragraph) as the following:

Social media (SM) has countless users worldwide, and the number is constantly growing, and the definition of SM varies (Lee & Lee, 2020). According to Obar and Wildman (2015), the definition of SM presented in the literature and the following commonalities among current social media services are: 1) Social media services are currently (currently) Web 2.0 Internet-based application, 2) User generated content is the lifeblood  of social media, 3) Individual and group creator user-specific profile for a sit or app designed and maintained by a social media services 4) Social media services facilitate the development of  social network online by connecting a profile with those of other individuals or/and groups. SM functionalities are not only traditionally designed for social networking purposes but have been widely used for business and work purposes. Hence, many available social media platforms are widely used by organizations for their official purposes, including Facebook, WeChat, DingTalk, WhatsApp, Twitter, Blogs, YouTube, and Photo-sharing sites (Pavithra & Deepak, 2020; Song et al., 2019).

3.        

 It will be important for the authors to elaborate further how the quota sampling was conducted. Also, there is a need to supplement more information regarding the inclusion and exclusion criteria of the sampling.

Thank you, reviewer 2

As suggested by reviewer 2, we have provided a better explanation regarding the quota sampling in 3.1.

We have added new subtopic 3.2 for the information regarding the inclusion and exclusion criteria of the sampling.

4.        

There is a need to mention how many missing data and how these missing data were treated in the current study.

Relevant paper:

Newman, D. A. (2014). Missing data: Five practical guidelines. Organizational Research Methods, 17(4), 372-411.

Thank you, reviewer 2

We have amended accordingly by adding a new subtopic, 4.1 data preparation, to explain missing data.

5.        

One limitation that I hope the authors can elaborate and discuss in their discussion is the possibility of reverse causation in the relationship between social media and stress. Given the use of cross-sectional design, it is still possible that stress is not a consequence but antecedent of social media use. This issue should be discussed and acknowledged and will help to provide a balanced perspective in this topic. See the following paper that will be relevant and helpful to the discussion:Does social media use increase depressive symptoms? A reverse causation perspective. (2021). Frontiers in Psychiatry12, 641934.

Thank you, reviewer 2

We have provided a better explanation in 7.0 Limitation and future work (last paragraph) as the following:

Lastly, future scholars should consider conducting a cross-sectional or longitudinal study by reversing causation in the relationship between social media and stress, especially in the workplace setting. This study has focused on the roles of SM toward technostress issues by discussing the consequences of SM use at work. To gain a clear picture of the SM-technostress link, the direction of social media causing psychological well-being problems might instead be reversed. As Hartanto et al. (2021) mentioned that future research needs to move toward a deeper analysis of the reverse possibility underlying social media use.

Reviewer 3 Report

Dear author.,

After reading carefully your manuscript, I noticed that you need to improve the logical flow of the study and many parts of the method, analysis, the research gap, the novelty of the research, and many more. You can refer to the manuscript for details of my comments and suggestions.

Thank you.

Author Response

Reviewer 3

No

Comments

Responds

1.        

This is the objective of the study?..

Thank you, reviewer 3

We have amended this issue.

2.        

Please stated the gap and inconsistency first before this sentence. I did not see the gap of research.... I suggest to add issue and gap of research.

Thank you, reviewer 3

We have amended by adding issues and gap of research in the abstract.

3.        

 Do you mean this employee for all sectors in Malaysia?.... better you stated 206 (goverment of non goverment sectors?...) employees

Thank you, reviewer 3

We have amended accordingly.

206 Malaysian employees from the government sector and government-linked companies (GLC) participated……….

4.        

Please add a conclusion after the result and policy implications.

Thank you, reviewer 3

We have provided a better explanation as the following:

…..This study provides theoretical and practical implications for extending the knowledge and provide mitigating plans and effort to resolve employees' performance at work. Therefore, this study fills the dearth of research pertaining to the roles of SM use at work on employees' innovative job performance.

5.        

Innovative Job Performance, please make it consistent the whole content of manuscript....

Thank you, reviewer 3

We have amended accordingly.

6.        

Highlight what the previous studies have done and what do you do in the current study.....

Thank you, reviewer 3

We have provided the following justifications:

Despite the existing body of knowledge, our understanding of SM overload is still constrained by some persisting gaps in the SM literature. The potential work-related consequences of SM stressors, especially in job performance, have remained understudied(Cho et al., 2019; Yu et al., 2018). Most prior studies on SM overload have been concentrated more on SM users generally (Cao et al., 2020; Dai et al., 2020; Fu et al., 2020), students (Guo et al., 2020; Islam et al., 2020) compared to the employees. Furthermore, studies on the different dimensions of overload remain scarce (Fu et al., 2020). Besides, in the context of SM, technostress associated with SM use has been studied primarily through the consequences of behavioural and psychological response (Bucher et al., 2013; Luqman et al., 2017; Maier, Laumer, Eckhardt, et al., 2015; Salo et al., 2019), and little attention to date focuses on the potential work-related outcomes such as job performance. Thus, this study provides a detailed process on how the association between SM use at work and SM overload can induce employees' psychological strain that interferes with employee's innovative job performance.

7.        

2. Literature Review

2.1 Underpinning Theory

2.1.1 xxxx

2.1.2 xxx

2.1.3 xxx

2.2 xxx

2.3 xxxx

 and so on....

2.2 Hypothesis development

Thank you, reviewer 3

We have renamed this section as suggested by reviewer 3.

8.        

Move to below figure caption and add Source:

Thank you, reviewer 3

We have amended accordingly in Figure 1.

9.        

Please writes the additional sentences to link both paragraphs

The interconnection of the

different SM overloads on employees' job performance remains limited.

Thank you, reviewer 3

We have provided the following justifications:

The interconnection of the different SM overloads on employees' job performance remains limited. Therefore, this study offers a significant extension of social media overload consisting of communication overload, information overload and social overload as representative stressors in understanding the psychological mechanisms underlying technostress on employees' innovative job performance.

10.     

Who said and year...

Pervasiveness SM usage has integrated in employees’ social life, socialization

and networking via SM within the organization.

Thank you, reviewer 3

We have cited the sentence as stated below:

Pervasiveness SM usage has integrated in employees’ social life, socialization and networking via SM within the organization (Song et al., 2019; Yang, 2020).

11.     

Do you mean this is a research framework...

Conceptual Framework?.....

Thank you, reviewer 3

We have renamed the framework.

12.     

You can provide this information in Table

Also, stated unit analysis and its measurement

Thank you, reviewer 3

We have amended accordingly by creating a new table for the list of the construct with the number of items and sources.

13.     

Descriptive statistics Right?.... You cannot run this by SmartPLS... means you have another statistical analysis besides SEM-PLS. please write up in methodology section

Thank you, reviewer 3

In the subtopic 3.5, we have provided the following justifications:

The data analysis began with the descriptive statistic, and this study performed SPSS version 22 to measure the frequency of background characteristics. Besides, this study applies the PLS-SEM approach……

14.     

Change in the form of Table and the caption is

Table 4: The result of the coefficient determination (R Square), Effect Size (f Square), predictive relevance (Q Square), and Impact of predictive relevance (q square).

You need to report and elaborate first about "coefficient determination (R Square), Effect Size (f Square), predictive relevance (Q Square) and Impact of predictive relevance (q square)." and threshold...

Thank you, reviewer 3

We have amended accordingly by creating a new table, Table 7 to report R2, f2, Q2 .

15.     

Discuss the result first and link it with findings of previous works

Thank you, reviewer 3

We have addressed all the comments and amended accordingly.

16.     

Research Implication can move and combine with section

6 Conclusion

6.1 Research Implication

6.2. Limitation and Future research

Thank you, reviewer 3

We have amended accordingly.

17.     

Please revise the content of this section and follow your RQ and RO.

Thank you, reviewer 3

We have amended accordingly.

Round 2

Reviewer 1 Report

I would like to congratulate the authors for the improvements. I have the following additional comments for the improvement of the paper:

1.) Despite the clarification of the definition of "social media" by the authors, the fact still stands that the participants in the study used "WhatsApp" during the study, and not various types of social media. Therefore, in my opinion the title of the paper, and the rest of the content, should reflect that fact. Instead of mentioning "social media" vaguely, they should specifically talk about WhatsApp usage. Any reflections to the effect of other social media usage (facebook, twitter, etc) on workers are in my opinion not possible. Each interface provides a different experience, and has a different effect on the end-users.

2.) In view of the revised statistical results (according to my comments in round 1), I believe that the fact that WhatsApp usage has a small effect on the variance of the Stressors (2.7% on social overload, 4.7% on comms overload and 5.5% on info overload), provides a very weak proof of the relevant hypotheses. Therefore,  the authors should not claim that "social media use at work positively influences information overload, communication overload, and social overload" in their discussion of findings, nor on the abstract and conclusion of the paper. They should instead explain that the effect of WhatsApp use at work on information overload, communication overload, and social overload seems to be very weak (2.7-5.5%). Changes need to be made accordingly throughout the paper.

Author Response

Kindly find the attachment for your perusal.

Thank you.

Reviewer 2 Report

The authors have addressed all my comments well. I appreciate all their efforts. Well done

Author Response

(The authors gave the same response as above.)

Reviewer 3 Report

Dear Author.,

Congratulations,

Your revised manuscript looks great and suitable to publish in this prestigious Journal. Please check carefully your variable in Figure e.g., Social Media. Also, your table captions. I suggest you use Mendeley or Endnote for managing the references.   

Author Response

(The authors gave the same response as above.)

Round 3

Reviewer 1 Report

It seems that although the authors have made some effort to revise the paper, they haven’t managed to cover the review comments from the previous round to a large extent. More specifically:

1.) My first comment in Round 2 was that the title of the paper, and the rest of the content, should reflect the fact that the users used WhatsApp during the study and not “social media” in general. Therefore, I asked from the authors to specifically talk about WhatsApp usage instead of mentioning "social media" vaguely. As each interface provides a different experience, and has a different effect on the end-users, the reader needs to understand that this study is on WhatsApp usage at work. This needs to be clear from the title of the paper to the end of the conclusion. Although the authors in their response letter claim to have covered this point, in essence they have not. I ask you once more to review all of the document and change the corresponding parts to reflect this fact (that this study is effectively mostly based on WhatsApp usage at work). Some more specific suggestions on this matter follow. At the same time, my second comment, that they need to change the strong allegations that SM usage strongly affects overload also was not correctly covered. There remain some incorrect and misleading statements in the paper. You may find more details in the next comments also.

2.) Although the title has been revised, it has not been changed correctly. In fact the new title is even more misleading than the previous title of the paper. It now looks as if the point of the paper is to prove that WhatApp is the dominant social media in workplaces. Moreover, the syntax is wrong and therefore the meaning is not conveyed properly. I urge the authors to change the title once more. Proposed new title: “The effect of WhatsApp social media usage on employee innovative performance at the workplace: Perspective from the Stressor-Strain-Outcome model”

3.) Although in their cover letter the authors claim that “Similarly, we have specified WhatsApp as the social media where necessary as the social media tool in this study”, I have found this not to be true in a number of places within the paper. For example in the middle of the Abstract the authors note (a) “…investigates how social media use at work can predict social media overloads…” . It should instead state that “…investigates how WhatsApp use at work can predict social media overloads…”, (b) “…Findings show that social media use at work has significantly influence information overload, communication overload, and social overload…”. The words “social media” should be changed to “social media – predominantly WhatsApp –” . Also, taking into account my comment from the previous review round that the statistics of the paper should be correctly reported throughout the paper, due to the very mild effect of SM usage on overload discovered in this study, the words “has significantly influence” should be changed to “has a mild, but statistically significant, effect on” in the Abstract.

4.) On Table 1 (page 8), the 3rd inclusion criterion is not reported correctly. It states “Use Facebook, Twitter, LinkedIn, WhatsApp, and Telegram”. However, based on the fact that the users were not screened to use ALL of the above, but either one of them, the word “and” should be changed to “or”. Moreover, based on the fact that WhatsApp was the SM used by the users in the end, I would place WhatsApp in the beginning of this sentence and not Facebook.

5.) On Figure 3, the left box “SOCIAL” should be changed to “Social Media use at work” just like what is depicted in the previous figure with the research model.

6.) In the first paragraph of the discussion, the authors claim that “These findings suggest that so-cial media use at work is central to stimulating social media -related overload”. However, based on the mild effect recorded in this study, I would change the words “is central to stimulating”, to “mildly, but statistically significantly, stimulates”.

7.) I suggest that Section 6.1 should be renamed from “Implication” (single and vague) to “Theoretical and Practical Implications” (plural and specific). Similarly, the title “Limitation and future work” should be changed to “Limitations and Suggestions for Future Work” in section 6.2.

8.) In the limitations, the authors need to add the fact that this study was performed with users that predominantly (almost exclusively) used WhatsApp as their social media during the study. Therefore, any findings ultimately reflect the usage of this specific SM. Hence future work must be made and research performed with the utilization of other kinds of social media at work (either combined usage, or exclusive usage of different SM platforms at work).More limitations must be recorded (and suggestions for future work accordingly), like the fact that the study was made on a specific country context, in a limited time frame (not inter-temporal), etc. Please expand both on your limitations as well as your suggestions for future work.

9.) I suggest that the paper is proof-read by an English native speaker, as it contains some  issues with expressions, grammar, etc. in various points.

Author Response

Dear Reviewer,

Kindly find the attachment for your perusal.

Thank you.
